# Comparative Pathology of Domestic Pigs and Wild Boar Infected with the Moderately Virulent African Swine Fever Virus Strain “Estonia 2014”

**DOI:** 10.3390/pathogens9080662

**Published:** 2020-08-16

**Authors:** Julia Sehl, Jutta Pikalo, Alexander Schäfer, Kati Franzke, Katrin Pannhorst, Ahmed Elnagar, Ulrike Blohm, Sandra Blome, Angele Breithaupt

**Affiliations:** Friedrich-Loeffler-Institut, 17493 Greifswald-Insel Riems, Germany; Julia.Sehl@fli.de (J.S.); Jutta.Pikalo@fli.de (J.P.); alexander.schaefer@fli.de (A.S.); kati.franzke@fli.de (K.F.); Katrin.Pannhorst@fli.de (K.P.); Ahmed.Elnagar@fli.de (A.E.); Ulrike.Blohm@fli.de (U.B.); Angele.Breithaupt@fli.de (A.B.)

**Keywords:** African swine fever virus, virulence, pathology, wild boar, domestic pig, macroscopy, histopathology, immunology

## Abstract

Endemically infected European wild boar are considered a major reservoir of African swine fever virus in Europe. While high lethality was observed in the majority of field cases, strains of moderate virulence occurred in the Baltic States. One of these, “Estonia 2014”, led to a higher number of clinically healthy, antibody-positive animals in the hunting bag of North-Eastern Estonia. Experimental characterization showed high virulence in wild boar but moderate virulence in domestic pigs. Putative pathogenic differences between wild boar and domestic pigs are unresolved and comparative pathological studies are limited. We here report on a kinetic experiment in both subspecies. Three animals each were euthanized at 4, 7, and 10 days post infection (dpi). Clinical data confirmed higher virulence in wild boar although macroscopy and viral genome load in blood and tissues were comparable in both subspecies. The percentage of viral antigen positive myeloid cells tested by flow cytometry did not differ significantly in most tissues. Only immunohistochemistry revealed consistently higher viral antigen loads in wild boar tissues in particular 7 dpi, whereas domestic pigs already eliminated the virus. The moderate virulence in domestic pigs could be explained by a more effective viral clearance.

## 1. Introduction

African swine fever (ASF) is a notifiable disease and one of the most important and serious threats to the pig industry today causing relevant global economic consequences [1]. The ASF virus (ASFV) belongs to the genus *Asfivirus* in the *Asfarviridae* family. The virus affects all species of the *Suidae* but only domestic pigs and Eurasian wild boar develop signs of a hemorrhagic fever like illness. Warthogs are considered reservoir hosts and do not display noticeable clinical signs. The same is probably true for other African wild suids. In Africa, the virus is endemic in sub-Saharan countries where it is vectored, especially in the sylvatic cycle with warthogs, by soft ticks of the genus *Ornithodorus* [2].

In 2007, ASF was introduced into Georgia and has then become a large-scale epidemic involving different European countries [3]. In 2018, the disease was detected in East Asia where it is progressively spreading [4,5,6].

Since (i) ASF established self-sustaining cycles, independent of ticks, and (ii) wild boar densities increased in the last decades, the wild boar is of particular importance in the spread of ASFV in Europe [7]. The ASFV strains circulating in Europe belong to the p72 genotype II, which are generally highly virulent in both domestic pigs and wild boar and cause acute disease with almost 100% lethality in animals of all ages and sexes [8,9]. Experimental infection reveals an incubation time of approximately 3–5 days. Typical clinical findings include high fever, dullness, and anorexia, but also comprise vomiting, bloody diarrhea, reddening of the skin, respiratory disorders, abortion, or stillbirth as well as neurological signs [5,10,11,12]. Animals die within 7–13 days pi commonly showing enlarged, hemorrhagic lymph nodes, reddening of tonsils, splenomegaly, petechial hemorrhages in different organs such as the kidney, colon, or urinary bladder as well as lung, and gall bladder wall edema (reviewed in [13]).

Subacute and chronic forms are also reported, which result from infection with moderately or low virulent virus strains. Infected animals can survive and even recover from the disease [14]. Such an attenuated phenotype was reported for an Estonian ASFV strain. In 2014, ASF has affected the Estonian wild boar population, and subsequently spread to domestic pig holdings [15]. Mortality and morbidity was reported to be variable: in the south of Estonia, mortality was high in wild boar while in contrast, in the northeast of the country, mortality was strikingly low and anti-ASFV antibodies were detected in hunted animals [16]. Domestic pigs in general showed only unspecific clinical signs, typical hemorrhagic fever was less frequently reported. Severe disease was primarily present in pregnant and nursing sows [15]. An initial animal trial with a north-eastern Estonia Isolate (Ida-Viru region) in European wild boar resulted in acute, severe disease and clinical signs typical for acute ASF. All but one boar succumbed to the infection. The survivor recovered completely, showed high antibody titers and did not establish a carrier state as evidenced by the lack of transmission to sentinels [16]. A follow-up characterization of the isolate obtained from the survivor was conducted in potbelly-type minipigs and domestic pigs. In total, 75% survived, showing transient infection with mild clinical signs [17]. In contrast, all wild boar, which were subsequently inoculated with an isolate obtained from a surviving domestic, died. In a further comparative trial with domestic pigs and wild boar (with predetermined days of necropsy), only three suckling wild boar piglets recovered while all domestic pigs showed only transient fever (Hühr et al., unpublished data).

Although to some extent inconsistent with field observations, there is cumulative evidence that the north-eastern Estonia isolate is moderately virulent and attenuated in domestic pigs, but still highly virulent in adult wild boar. Genome sequence analysis revealed a 14.5 kilobase pair deletion at the 5’end of viral DNA, which is responsible for the attenuated phenotype in domestic pigs [17]. Comparative pathological studies on ASFV isolates in domestic pigs and wild boar are limited on precise, systematic and semi-quantitative evaluations of macroscopic and/or histopathologic lesions [18] as well as target cell identification. Thus, information on host factors for virulence is scarce.

We therefore aimed to compare the early pathogenesis of ASFV “Estonia 2014” in domestic pigs and wild boar. In a kinetic study, we investigated the clinical outcome for correlation with the macroscopic and histopathologic lesion, the viral antigen distribution and the cellular response to ASFV infection. For this purpose, we inoculated nine domestic pigs and nine wild boar with ASFV “Estonia 2014” [17] and monitored daily for clinical signs using a harmonized scoring system. Three animals each were euthanized and analyzed at 4, 7, and 10 days post infection (dpi).

## 2. Results

### 2.1. Clinical Disease

Clinical scores of animals were assessed based on the protocol published by Pietschmann et al. (2015). Following oronasal inoculation, the incubation period was 4 days in both subspecies. The animals showed general depression, anorexia, curved back, ataxia, respiratory distress, and increased recumbency. In brief, wild boar revealed higher clinical scores (up to score 10.5) than domestic pigs (up to score 5) and remaining domestic pigs recovered until day 10 (score 0), whereas wild boar still presented with apparent moderate disease (score 6–7) (Figure 1).

In detail, the clinical score of 5/12 wild boar (animals #13, #15, #17, #27, #28) was affected by signs not related to the infection from the very beginning. Hierarchic encounters led to lameness, slightly affected liveliness, posture, breathing, and laying (Figure 1A). These animals were evaluated with maximum 4 points before ASF became clinically apparent. Starting at 4 dpi, clinical signs typical for ASF were noted (animals #13, #15, #20, #27, #28) and affected all animals from day 5 onwards. The maximum clinical score of 10.5 was reached at 7 dpi (wild boar #28). The wild boar at 10 dpi had clinical scores from 6 to 7 points.

At 4 dpi, domestic pig #30 showed diarrhea (score 1), all other pigs were normal (Figure 1B). The clinical score in domestic pigs increased from day 4 to day 7 pi to maximum 5 points mainly affecting liveliness, position, breathing, feed intake, and walk. Until day 10 pi all remaining domestic pigs clinically recovered. Additionally, body temperature was measured daily in domestic pigs, fever (>40.0 °C) was recorded starting at day 6 pi, lasting until day 10 with up to 40.6 °C. Temperature profiles from 0 to 10 dpi of individual domestic pigs are shown in Appendix A. Domestic pigs and wild boar, which served as control animals, were clinically healthy.

### 2.2. Viral Genome Load in Blood and Tissues

Prior to inoculation, all animals were tested negative for ASF viral antigen, viral genome, and virus using lateral flow assays (LFD), ASFV-specific qPCR, and hemadsorption test (HAT), respectively. The back titration of the inoculated virus calculated according to Spearman and Kärber [19,20] verified the administered titer of 1 × 10^5.25^ hemadsorbing units (HAU) per mL per pig. 

The viral genome load in blood and tissue samples was tested by qPCR for correlation with the clinical course (Figure 2). Blood cells were further investigated by flow cytometry and the frequency of p72^+^ myeloid subsets was determined. In correlation with the onset of clinical disease, ASFV genome was detectable from day 4 pi onwards in blood and tissues. The genome load in the blood peaked with comparable amounts in both subspecies on day 7 (wild boar: 10^4.74^ genome copies per µL, domestic pigs: 10^4.35^ genome copies per µL) and only slightly declined on day 10 pi (wild boar: 10^4.41^ genome copies per µL, domestic pigs: 10^3.95^ genome copies per µL). The highest ASFV genome loads in tissues were detected in the spleen (10^2.8^–10^4.9^ genome copies per µL), followed by the liver (10^1.9^–10^4.2^ genome copies per µL) and lung (10^1^–10^4.1^ genome copies per µL) and much lesser in the inguinal lymph node (10^0^–10^3.4^ genome copies per µL). Viral genome loads did not differ significantly at 4 dpi between wild boar and domestic pigs, but were slightly higher in wild boar compared to domestic pigs at 7 dpi. Although the ASFV genome loads decreased at 10 dpi in all tissues of both subspecies, it was still above 10^3^ genome copies per µL. At this time wild boar revealed significantly higher viral genome loads in the blood and liver.

Flow cytometric analyses of blood samples showed, that the relative proportions of FSC^hi^/SSC^hi^/CD172^+^/CD14^+^ cells (granulocytes), FSC^med^/SSC^low^/CD172^+/^CD14^+^ cells (monocytes), and FSC^med^/SSC^low^/CD172^+^/CD14^–^ cells (dendritic cells) stayed within the range seen in control animals throughout the experiment. The proportion of p72^+^ cells did not differ significantly between wild boar and domestic pigs. The relative amount of p72^+^ monocytes and dendritic cells was below 20% at day 4 pi in both subspecies, slightly increased at day 7 pi but markedly peaked at day 10 pi with up to 90% of monocytes and approximately 40% of dendritic cells affected. There were no significant differences in the p72 mean fluorescence intensity (MFI) in both subspecies at day 10 pi (wild boar: DC 737 ± 68, monocytes 887 ± 38; domestic pig: DC 730 ± 132, monocytes 819 ± 107). The percentage of p72^+^ granulocytes was high from the beginning in wild boar (40%) and domestic pigs (60%) and continuously increased until day 10 pi (up to 98% in both subspecies). The p72 MFI was higher in granulocytes compared to other myeloid populations but did not differ significantly between the two subspecies (wild boar: 1221 ± 67; domestic pig: 1408 ± 163).

### 2.3. Gross Pathology

Full autopsy was conducted on all animals. Standardized gross scoring was performed on the following organs based on Galindo-Cardiel et al. [21]: tonsils, spleen, lymph nodes (hepatogastric, renal and popliteal), lung (cranial and caudal lobes), liver with gall bladder, and kidneys. Gross lesions were generally mild to moderate and were detected in wild boar and domestic pigs from day 4 pi on. The severity of lesions increased over time in both subspecies but did not differ markedly.

Slightly enlarged, hemorrhagic lymph nodes were present on 4 dpi in 2/3 wild boar but not in domestic pigs. At 7 dpi 2/3 domestic pigs began to show mild to moderate enlargement of lymph nodes whereas only 1/3 wild boar was affected. At day 10 pi all domestic pigs and 2/3 wild boar showed mainly moderate enlargement. Lymph node hemorrhages, particularly affecting the hepatogastric and renal lymph node, were found in 1/3 wild boar at day 4 pi. All domestic pigs had hemorrhages at day 7. At 10 dpi, 2/3 wild boar and all domestic pigs showed hemorrhages. Besides mild enlargement, mild hemorrhages were only rarely observed in the popliteal lymph node over the study period (4/9 wild boar, 4/9 domestic pig).

Moderate petechiae (cortico-medullar pattern) mainly affecting the renal cortex occurred at day 10 pi in all wild boar and in 2/3 domestic pigs. Gross appearance was largely comparable in both subspecies even though the amount of petechiae varied.

In the lung, up to moderate multifocal to coalescing consolidated areas were observed in 2/3 domestic pigs at 7 dpi, mainly affecting the cranial lobes (Figure 3A). On day 10 pi all domestic pigs revealed pulmonary lesions while only one wild boar showed a small consolidated area in the cranial lobe (Figure 3B).

Occasional findings included moderate diffuse hemorrhagic gastritis in a wild boar at day 7 pi, edema of the gall bladder wall in a domestic pig at 10 dpi, severe multifocal myocardial necrosis and hemorrhage in a domestic pig on day 10, and intratonsillar abscesses in wild boar at day 7 and 10 pi. All findings were confirmed by histopathology. No lesions were found in the spleen. Macroscopically, no changes were observed in the control animals.

### 2.4. Histopathological, Immunological, and Electron Microscopical Analysis

From days 4, 7, and 10 pi hematoxylin-eosin stained sections of the spleen, hepatogastric and popliteal lymph node, palatine tonsil, bone marrow, lung, liver with gall bladder, kidney, cerebrum, and cerebellum were analyzed and scored semi-quantitatively based on Galindo-Cardiel et al. [21]. T-cell subsets of the immune cell infiltrates detected in the spleen, lung, liver, and hepatogastric lymph node were further investigated by flow cytometry. Additionally, immunohistochemistry was performed to identify and semi-quantify p72-positive target cells and cells undergoing apoptosis. To gain deeper insights in putative target cells within the myelomonocytic cell lineage, the presence of the ASFV protein p72 was tested in myeloid cells by flow cytometry. In addition, representative tissue sections of the spleen, lung, and liver from wild boar and domestic pigs were investigated for cell-specific presence of ASFV particles by electron microscopy at indicated time points.

#### 2.4.1. Spleen

Although splenic gross lesions, like splenomegaly or infarction, were not evident, histopathological changes were present (Figure 4). Whereas hematoxylin eosin-staining revealed no specific differences between wild boar and domestic pigs, immunohistochemistry yielded markedly different results (Figure 5A,B). In detail, apoptosis of lymphocytes in the white pulp was up to moderately present in control animals as well as at 4 dpi. Apoptosis slightly increased at day 7 and became less at day 10 (Figure 4A). Apoptosis of lymphoid cells was confirmed by active caspase-3 labelling (Figure 4B, see immunohistochemistry in the inset). In correlation with the onset of clinical disease, apoptosis and necrosis of myelomonocytic cells in the red pulp were mild at 4 dpi, mild to severe at 7 dpi, and mild to moderate at day 10 (Figure 4C). Additionally, myelomonocytic cells of the red pulp appeared slightly swollen (hypertrophic) at day 4 in single wild boar and domestic pigs and became more prevalent at 7 and 10 dpi in wild boar and domestic pigs (Figure 4D, inset). Immunohistochemistry identified viral antigen abundantly (score 3) in myelomonocytic cells, in particular in the red pulp at day 4 pi in all animals (Figure 5A). At day 7 pi a high amount (score 3) of positive cells was still detectable in wild boar while viral antigen load markedly decreased in domestic pigs (score 0–2). At day 10 pi scattered positive cells (score 1) were detectable in one domestic pig. ASFV p72 antigen detection is illustrated in Figure 5B. 

In contrast to p72 antigen detection by immunohistochemistry, flow cytometric analysis showed that the proportion of infected CD172^+^/CD14^+^ monocytes was moderate at 10 dpi in both species (mean 30–40%) while there were only few p72^+^ CD172^+^/CD14^-^ dendritic cells in the spleen (Figure 5C). Electron microscopy confirmed infection of monocytes/macrophages showing abundant vacuoles and intracytoplasmic ASFV particles (details not shown).

#### 2.4.2. Lymph Nodes

Lymphocytolysis (syn. apoptosis) in lymphoid follicles of the hepatogastric as well as popliteal lymph node, was mildly present in control animals. Mild to moderate lymphoid apoptosis was consistently present in both lymph nodes of both subspecies, except for one wild boar showing severe lymphocytolysis in the hepatogastric lymph node. Apoptosis and necrosis, affecting the perifollicular cortex and paracortex was found in infected animals at varying degree, starting at 4 dpi. The hepatogastric lymph node was generally more consistently and more severely affected compared to the peripheral, popliteal lymph node (details not shown).

Up to moderate ASFV antigen was detected at day 4 pi in single wild boar and domestic pig in both lymph nodes (Figure 6A,C). Particularly in wild boar, the amount of p72^+^ cells increased markedly but was low in domestic pigs by day 7 pi, shown in Figure 6B,D. At day 10 pi, viral antigen was undetectable in hepatogastric lymph nodes or only slightly found in single animals in the popliteal lymph node. Flow cytometric analysis yielded only few monocytes and dendritic cells in the hepatogastric lymph node from which primarily monocytes were p72-positive (details not shown).

Further, histopathology confirmed macroscopically recorded hemorrhages in the hepatogastric and popliteal lymph nodes, and additionally congestion was found.

#### 2.4.3. Palatine Tonsil

Apoptosis of lymphoid follicle cells was consistently detected in infected but also in control animals. Mild congestion and only very mild necrosis was observed at 7 dpi in two wild boar and one domestic pig. In all infected and non-infected animals, tonsillary crypts were filled with variable amount of viable and degenerate neutrophilic granulocytes. This was particularly pronounced in two wild boar from day 7 and one wild boar from day 10 pi. The crypt epithelium as well as myelomonocytic cells were slightly positive for ASFV antigen in one domestic pig at 4 dpi. All infected animals from day 7 pi revealed a high amount (score 1–3) of p72 antigen in the tonsil, most abundantly in wild boar. In one wild boar, the mucosal epithelium was ASFV antigen positive. At day 10 pi only one wild boar but all domestic pigs showed scattered p72-labelling in the crypt epithelium and myelomonocytic cells (details not shown).

#### 2.4.4. Bone Marrow

The bone marrow revealed no histopathological changes, but immunohistochemistry showed p72-positive myeloid cells, including megakaryocytes. High amounts of viral antigen were detected in all wild boar at day 7 pi while all domestic pigs showed fewer positive cells. At day 10 pi, only few positive cells were present in two wild boar (details not shown).

#### 2.4.5. Lung

Histopathology identified interstitial inflammation in the lungs, mainly affecting the cranial lung lobes, characterized by infiltrating lymphocytes and few macrophages. Detailed evaluation revealed for wild boar mild to moderate interstitial pneumonia at 4 dpi in 2/3 animals, and in all wild boar at 7 dpi. No inflammation was detectable at 10 dpi. Domestic pigs also exhibited mild pneumonia at 4 dpi in 2/3 animals, and mild to moderate inflammation at 7 dpi. In contrast to wild boar, moderate to severe pneumonia was found at 10 dpi (Figure 7A,B). In addition, interstitial macrophages appeared more prominent in single animals at day 4 pi, which particularly increased in wild boar at day 7 pi and was present until the end of the study in both groups. Of note, mild infiltrates were also found in one control wild boar and domestic pig. Mild alveolar edema was only occasionally detected, but was severe in wild boar #28 euthanized at day 7 pi with the highest clinical score (score 10.5). Further unspecific findings included pulmonary congestion, found in the majority of controls and infected animals.

To identify the phenotype of infiltrating cells and compare the responses in wild boar and domestic pigs, flow cytometric analysis was performed on control and day 10 lung tissues. The overall response at day 10 was comparable in both subspecies. We detected a significant increase of CD8α^+^ αβ T cells in both subspecies. Moreover, CD4^+^/CD8α^+^ αβ T cells increased significantly in wild boar and domestic pigs but considerably less pronounced, while CD4^+^ αβ T cells showed a corresponding decrease (Figure 7C). The frequencies of activated γδ T cells were significantly elevated in wild boar but showed only a small increase in domestic pigs. Effector γδ T-cell frequencies were not altered in both subspecies (Figure 7C).

By immunohistochemistry, ASFV antigen was detected multifocally in mononuclear cells in all animals starting at day 4 pi, where the number of positive cells was comparably low in wild boar and domestic pigs (Figure 8A). At day 7 pi viral antigen was found abundantly (score 3) in all wild boar whereas all lungs of domestic pigs revealed markedly lower amounts (score 1) of positive cells. At day 10 pi antigen detection yielded similar scores for wild boar and domestic pigs (score 1–2). Figure 8B shows representative anti-p72 labelled lung sections from day 4 to 10 pi. Immunological investigation demonstrated that at 10 dpi most CD172^+^/CD14^+^ monocytes in the lungs were p72-positive, while only around 50% of the dendritic cells were positive for viral p72. Moreover, there were no changes in the frequencies of lung monocytes in both subspecies. Of note, wild boar showed significantly higher dendritic cell frequencies than domestic pigs 10 dpi (Figure 8C). By electron microscopy, ASFV particles were identified in pulmonary intravascular macrophages/monocytes (PIM) (Figure 9).

#### 2.4.6. Liver

Histopathology identified lymphoid infiltrates mainly affecting the hepatic sinuses, increasing over time in both subspecies (Figure 10A,B). Starting on 4 dpi in 2/3 domestic pigs, sinusoidal infiltrates were consistently present in wild boar and domestic pigs at 7 (mild to moderate) and 10 dpi (moderate). Flow cytometry revealed infiltrating lymphocytes to be mainly CD8α^+^ and to a lesser extent also CD4^+^/CD8α^+^ αβ T cells, which increased significantly in wild boar and domestic pigs compared to controls. CD2^+^/CD8^–^γδ T-cell frequencies were also elevated in both subspecies (Figure 10C). In addition, microscopic investigation found Kupffer cell degeneration, characterized by swelling and detachment, as well as necrosis. Starting mildly at day 4 pi in one domestic pig, degeneration and necrosis was found more frequently and more severe in wild boar compared to domestic pigs and lasted until day 10 pi (Figure 11A,B). The degeneration of Kupffer cells correlated with a clear drop of CD172^low^/CD14^+^ cells in the liver at day 10 pi in wild boar and domestic pigs, as found in the flow cytometric analyses (Figure 11C). Immunohistochemistry demonstrated viral antigen in all wild boar and domestic pigs at 4 dpi mainly in sinusoid-lining cells, indicative for Kupffer cells and/or endothelium (score 1–3) but also in few hepatocytes until day 7 pi (data not shown). Whereas all wild boar exhibited abundant (score 3) viral antigen, 2/3 domestic pigs showed only few positive sinusoid lining cells (score 1) at 7 dpi. No p72^+^ cells were detected at 10 dpi in both subspecies (Figure 12A). Flow cytometry showed increasing proportions of p72^+^ CD172^+^/CD14^+^ monocytes (up to 70%), CD172^+^/CD14^−^ dendritic cells (up to 40%), and CD172^low^/CD14^+^ putative Kupffer cells (up to 20%) over time. We found higher amounts of p72^+^ myeloid cells in wild boar compared to domestic pigs, especially given the fact that the frequencies of monocytes and dendritic cells were significantly higher in wild boar than in domestic pigs 10 dpi (Figure 12B). However, given that there were only few Kupffer cells left, the majority of p72^+^ cells consisted of monocytes. The infection of Kupffer cells was also confirmed by electron microscopy (Figure 12C). Congestion of the liver was detected in all infected and control animals, but was slightly more severe at day 4 pi.

#### 2.4.7. Kidney

In addition to macroscopically recorded petechial hemorrhages, congestion of the cortex and medulla were observed in nearly all of the infected, but also in the control animals. Mild interstitial lymphocytic infiltration confined to the cortex was found in one control wild boar and in the majority of infected wild boar. ASFV antigen was up to moderately detected in the cortical and medullary interstitium at day 7 in 3/3 wild boar and 1/3 domestic pigs, and mildly at 10 dpi in 3/3 wild boar only (details not shown).

#### 2.4.8. Brain

Sections of the cerebrum at the level of the hippocampus revealed no histopathological changes, but viral antigen was detected in few cortical glial cells in one wild boar at day 7 pi, in two wild boar and one domestic pig at day 10.

The cerebellar section was evaluated at the level of the pons. Although histopathological changes were absent, immunohistochemistry revealed focal antigen positive cells in the granule cell layer and choroid plexus in one wild boar at day 7 pi and positive glial cells in one wild boar at day 10 pi (details not shown).

## 3. Discussion

Differences in the disease course and in susceptibility to ASFV infection between domestic pigs and European wild boar have been described, indicating differences in the pathogenic mechanisms in the subspecies [18] but mostly exclude histopathology and target cell identification. In the present study, we compared domestic pigs and wild boar infected with ASFV “Estonia 2014” that previously showed an attenuated phenotype in domestic pigs but still high virulence in adult wild boar [17]. To gain more insights on host factors for virulence, we analyzed three domestic pigs and three wild boar each on 4, 7, and 10 dpi in a kinetic trial. We scored clinical signs, gross pathological changes and histological lesions, as well as antigen distribution by immunohistochemistry. Blood and tissue samples were examined for viral genome load by qPCR. Electron microscopy as well as flow cytometry were performed to further evaluate changes observed pathomorphologically.

Clinical scoring revealed ASF-related disease starting 4 dpi. Wild boar showed much higher clinical scores than domestic pigs. Whereas domestic pigs fully recovered until day 10, wild boar still exhibited moderate disease. This mild clinical picture and subsequent recovery of domestic pigs reflects experimental data obtained by Zani et al. [17] with this virus strain. The very same study confirmed a more severe clinical course in wild boar, with two female 2-year old wild boar found dead at 8 and 9 dpi and 6–8-month-old piglets were euthanized at days 16 and 17 pi. This indicates that virulence might be age-related, although the number of animals used was low and it is generally assumed that age-dependence of clinical signs does not exist for highly virulent ASFV strains [22]. Since our study focused on early lesions and viral antigen distribution and thus lasted 10 days only, we cannot exclude that the wild boar might have reached the humane endpoint later on or could have recovered completely.

ASFV genome was detectable from day 4 pi onwards in blood and tissue samples. The viral genome load in tissues is most likely dependent on the viral load in the blood since the values for both were comparably high over the study period. However, no correlation with the overall clinical course was found based on (i) the lack of marked pathological differences between wild boar and domestic pigs, in particular at 10 dpi, (ii) absence of clinical disease at 10 dpi in domestic pigs with persistent high viral loads in blood and tissues with more than 10^3^ genome copies per µL, and (iii) almost comparable genome loads in tissues of wild boar at days 4 and 10 pi but a more severe disease at 10 dpi. Flow cytometric analyses of whole blood samples showed that the relative proportions of granulocytes, monocytes, and dendritic cells did not change throughout the experiment and that frequencies of p72^+^ cells did not differ between wild boar and domestic pigs. In line with previous observations in granulocytes of domestic pigs inoculated with highly virulent ASFV [23] viral antigen was abundantly found in our study. It remains unclear whether the viral antigen in granulocytes derives from productive infection, as it has been shown for highly virulent ASFV strains [23], or phagocytosis of cell debris and erythrocytes with viral attachments, or a combination of both. Given that erythrocytes carry the vast majority of infectious particles in the blood [24], it seems likely that cells with high phagocytosis activity and possible viral replication, i.e., granulocytes, have a higher MFI than other myeloid cell subsets. Moreover, since granulocyte identification was conducted by FSC vs. SSC gating, we cannot rule out that large macrophages might have been found in the granulocyte gate. Given that up to 50% of all live blood leukocytes were granulocytes and most of them were positive for the viral protein p72, the major antigen load in the leukocyte fraction of the blood was found in granulocytes throughout the whole study period. We also provide evidence that cells of the myeloid lineage are affected by ASFV infection in various tissues. It has been shown that ASFV is able to infect and modulate dendritic cells in vitro [25,26], however, in vivo evidence was scarce [27,28]. The ASFV protein p72 was found in dendritic cells in virtually all tissues investigated. ASFV infection of dendritic cells is thought to impair their function [29] but this requires further investigation.

The role and impact of neutralizing antibodies upon ASFV infection are still controversially discussed, and full neutralization could not be shown in recent studies with convalescent animals (Petrov et al. 2018). However, the production of efficiently neutralizing antibodies in domestic pigs could have influenced the disease outcome and explain convalescence in domestic pigs in our study as it has been described after infection with moderately virulent strains [30].

Although ASFV is an obligate pathogen for both wild boar and domestic pigs, potential genetic differences among and across the two subspecies should be considered as they could influence the susceptibility to ASFV and thus disease outcome [31]. Different clinical signs, immunological responses, and pathological outcomes are also known for highly pathogenic porcine reproductive and respiratory syndrome virus infection in wild boar and domestic pigs [32]. So far, genetic discrimination between domestic pigs and wild boar has been challenging since there have been only few genetic markers identified to distinguish between domestic pigs and wild boar [33], and even immunogenetic investigation of domestic pigs and wild boar could not identify any statistically significant differences in allele frequency and heterozygosity across SNPs [34]. However, very recently SLA-1 diversity was investigated in different domestic pig breeds showing significant distribution of SLA-1 alleles among pig breeds which could explain differences in the immune response [35]. This has not been done for wild boar so far and underlines the necessity of further investigation.

In accordance with earlier studies, macroscopic changes did not generally differ between domestic pigs and wild boar [36]. At 4 dpi, changes were generally mild and gradually became more severe until 10 dpi. In particular, typical hemorrhages in lymph nodes and kidneys were found as already described [16,17].

Apoptosis of lymphoid cells (syn: lymphocytolysis) could be confirmed for the spleen and lymph nodes [37,38,39,40,41] but did not differ between wild boar and domestic pigs. Apoptosis/necrosis in the splenic red pulp as well as in perifollicular and paracortical regions of lymph nodes were also present to the same extent in wild boar and domestic pigs, though not as severe as after infection with highly-virulent strains found in the spleen [42]. Immunohistochemistry revealed much more viral antigen positive cells in the splenic red pulp as well as in the lymph nodes at day 7 pi in wild boar compared to domestic pigs, indicating differences in the pace of viral clearance. Higher numbers of viral antigen positive cells in wild boar were also present in the palatine tonsil, although pathohistological changes were rather mild and largely similar in domestic pigs and wild boar.

Pulmonary consolidation was present in almost all domestic pigs at day 7 and 10 pi (up to 70% affected lung lobe), but only in one wild boar (<5% affected). Histopathology identified up to severe lymphohistiocytic interstitial pneumonia, partially also affecting macroscopically normal lungs of both subspecies. Pneumonic lesions were particularly associated with viral antigen labelled cells but those were also found disseminated in otherwise unaffected lung tissue. At day 7 pi viral antigen was found abundantly in wild boar, but only minimally in domestic pigs, again indicating an impaired virus elimination in wild boar. Pneumonia is described in domestic pigs after field and experimental infection [43,44] and can be either associated with ASF or secondary infections [45]. Supporting the histopathologic findings, flow cytometry of T cell subsets in lung tissue showed an immune response mainly driven by CD8α^+^ αβ T cells and CD2^+^/CD8α^–^ activated γδ T cells. A comparable T-cell response was also found in the liver. CD8α^+^ T cells are known to be pivotal mediators of the anti-ASFV response [46]. Recently, it has been shown that highly virulent ASFV infections induce different responses in wild boar and domestic pigs [47]. Wild boar mounted a cytotoxic Th1 response, while the T-cell response in domestic pigs seemed to be impaired. Neither response was beneficial as all animals showed signs of severe disease and most succumbed to the infection [47]. During the moderately virulent ASFV infection in the present study, the pulmonary and hepatic inflammations seem to be driven mainly by cytotoxic CD8α^+^ αβ T cells in both subspecies. A distinct response of cytotoxic T cells might cause the tissue damage and cell degeneration observed in this study, indicating a possible role for immunopathologic processes in ASF pathogenesis which urges further research.

We also observed an increase in the number and size of pulmonary macrophages which is fully consistent with the findings shown by Carrasco et al. [48] who found proliferation particularly of pulmonary intravascular macrophages (PIM) in response to intracytoplasmic immunocomplexes after ASFV infection. In the present study, the infection of PIM was demonstrated in wild boar by electron microscopy, one of the main target cells of ASF in the lung of domestic pigs [49,50].

In the liver, lymphocytic infiltrates were initially observed in the sinuses only in domestic pigs, but the values in domestic pigs and wild boar increased and converged until day 10. Since p72 antigen was no longer detectable with immunohistochemistry at day 7 pi in domestic pigs, but was still present in wild boar, again ASFV clearing seems to be more effective in domestic pigs. Kupffer cell degeneration, necrosis, and loss was obvious in both subspecies although slightly more severe in wild boar. This finding is typical for ASF as reported by Carrasco et al. [48] and correlated with flow cytometry results in the present study, showing a clear drop of CD172^low^/CD14^+^ cells beginning day 7 pi in wild boar and domestic pigs.

Higher numbers of viral antigen positive cells in wild boar were also present in the kidney and brain although pathohistological changes were rather mild and largely similar in domestic pigs and wild boar.

In summary, we confirmed that ASFV “Estonia 2014” is more virulent in wild boar than in domestic pigs. However, autopsy as well as routine histopathology failed to identify significant differences in the lesion profile and could not identify the cause for the more pronounced disease in wild boar. Further, the viral genome load in tissues is most likely attributed to the viral load in the blood and is thus not suitable to evaluate affected target tissue and cells. The viral genome load also remains almost unchanged over time, again not reflecting the clinical course. Using flow cytometry, we could show that cells of the myeloid lineage, including monocytes, dendritic cells, granulocytes, and myeloid cells in the liver, contain viral antigen in various tissues. Even though we could not find distinct differences in target cells or virus tropism between wild boar and domestic pigs, immunohistochemistry showed striking differences at 7 dpi: while antigen detection in wild boar peaked at day 7 in main target tissues, i.e., spleen, liver, and lung, the domestic pigs have already started to eliminate the virus. This raises the question of whether in the early phase of the disease, the delayed viral clearing in wild boar is relevant for the virulence of ASFV Estonia. To address this, particularly immunologic-mechanistical and functional insights will be important to shed light in this field. In this respect, comparative cytokine and large-scale immune cell profile analysis in domestic pigs and wild boar are needed to explain the different phenotypes after infection with ASFV “Estonia 2014”.

## 4. Materials and Methods

### 4.1. Study Design

In the animal experiment, all applicable animal welfare regulations including EU Directive 2010/63/EC and institutional guidelines were taken into consideration. The animal experiment was approved by the competent authority (Landesamt für Landwirtschaft, Lebensmittelsicherheit und Fischerei (LALLF) Mecklenburg-Vorpommern) under reference number 7221.3-2-011/19.

The present study was carried out for detailed pathological analysis of disease dynamics of domestic pigs and wild boar infected with an attenuated Estonian ASFV isolate (“Estonia 2014”). To this end, an ASFV-positive macrophage culture supernatant was prepared to a final titer of approximately 1 × 10^5.25^ hemadsorbing units (HAU) per mL. Domestic pigs and wild boar were inoculated oro-nasally. The dilutions were based on an end-point virus titration of the original material on macrophages derived from peripheral blood monocytic cells (PBMCs). Upon application, back titration was carried out to confirm the administered virus-containing dose.

### 4.2. Cells

For the preparation of PBMC-derived macrophages blood was collected from healthy domestic donor pigs that are kept in the quarantine stable at the Friedrich-Loeffler-Institut (FLI). Briefly, PBMCs were obtained from EDTA-anticoagulated blood using Pancoll animal density gradient medium (PAN Biotech, Aidenbach, Germany). PBMCs were grown in RPMI-1640 cell culture medium with 4-(2-hydroxyethyl)-1-piperazineethanesulfonic acid (HEPES) and 10% fetal calf serum (FCS) at 37 °C in a humidified atmosphere containing 5% CO_2_. The medium was supplied with amphotericin B, streptomycin, and penicillin to avoid bacterial and fungal growth. To facilitate maturation of macrophages, GM-CSF (granulocyte macrophage colony-stimulating factor; Biomol, Hamburg, Germany) was added to the cell culture medium with a concentration at 2 ng/mL.

### 4.3. Virus

The Estonian virus “Estonia 2014” isolate belongs to genotype II with a deletion at the 5’ end (GenBank Accession number LS478113.1) that corresponds to an attenuated phenotype [17]. The virus was grown on PBMCs and the supernatant was used for the animal trial with a final titer of 1 × 10^5.25^ HAU per mL. The titer was confirmed by back titration of the inoculum.

### 4.4. Animal Experiment

The study included 12 6–8-month-old European wild boar and 11 10-week-old domestic pigs of mixed sex. Wild boar were obtained from three different wildlife parks, domestic pigs from one commercial fattening farm. The animals were housed in groups in the high containment facility of the FLI (L3+) prior to infection. All animals were individually ear-tagged. The animals were fed a commercial pig food with corn and hay-cob supplement and had access to water ad libitum. After an acclimatization period of 1 week, the animals were inoculated oro-nasally with 2 mL of 1 × 10^5.25^ HAU per mL. On day 0, three wild boar and two domestic pigs were euthanized and used as control animals. On days 4, 7, and 10, three animals from each group (domestic pigs and wild boar) were euthanized and submitted to necropsy. The animals were deeply anaesthetized with tiletamine/zolazepam (Zoletil^®^, Virbac, Carros cedex, France), ketamin (Ketamin 10%, Medistar, Ascheberg, Germany) and xylazine (Xylavet 2%, CP Pharma, Burgdorf, Germany) and subsequently killed by bleeding.

#### 4.4.1. Clinical Scoring

Clinical parameters of all animals were assessed daily based on a harmonized scoring system as previously described [51]. The sum of the points was recorded as the clinical score (CS) that was also used to define humane endpoints prior to the experiment. The body temperature was measured daily in the domestic pigs and at day 10 pi in wild boar. Blood samples were collected at day 0 before inoculation and before euthanasia on days 4, 7, and 10 post infection.

#### 4.4.2. Tissue Sample Collection

Full autopsy was performed on all animals and tissue samples including the spleen, hepatogastric and popliteal lymph nodes, palatine tonsil, bone marrow, lung (cranial and caudal lobe), liver with gall bladder, kidney, and brain (cerebellum and cerebrum) were taken for histopathological investigation. For electron microscopy, lung, spleen, and liver were collected. Blood (EDTA, serum, citrate) and swab samples (cotton swab, genotube, prime store) were taken for further analysis (data not shown).

#### 4.4.3. Pathomorphological and Electron Microscopical Analysis

##### Gross Pathology and Macroscopic Scoring

During necropsy all animals were scored based to the gross protocol published by Galindo-Cardiel et al. [21] with slight modifications as follows. Organ lesions were generally scored on an ordinal scale ranging from 0 to 3 with normal (0), mild (1), moderate (2), or severe (3). In addition, lymphatic tissue including the spleen, palatine tonsil, and lymph nodes were examined for hyperplasia (size increased), hemorrhage, and necrosis and scored accordingly. Hemorrhages in the kidney were evaluated with normal (0), petechiae (1), ecchymoses (2), and diffuse hemorrhage (3). The percentage distribution of pulmonary consolidation was determined for *the lobus cranialis sinister pars cranialis, lobus cranialis sinister pars caudalis, lobus caudalis sinister, lobus cranialis dexter, lobus medius, lobus caudalis dexter and lobus accessorius*. Tissue samples were fixed in 10% neutral-buffered formalin for at least 3 weeks.

##### Histopathology and Immunohistochemistry

Representative sections of each fixed organ sample were cut and embedded in paraffin wax. Bone tissue was decalcified for at least 3 days in Formical 2000 (Decal, Tallman, New York, NY, USA). Embedded sections were cut at 2–3 µm thick slices, mounted on Super-Frost-Plus-slides (Carl Roth GmbH, Karlsruhe, Germany), and stained with hematoxylin-eosin.

For immunohistochemistry, paraffin-embedded sections were dewaxed and rehydrated through ascending concentrations of ethyl alcohol. Sections were rinsed with deionized water (A. dest.) and treated with 3% of hydrogen peroxide (Merck, Darmstadt, Germany) for 10 min to block intrinsic peroxidases. After washing with A. dest. sections were demasked with Tris/EDTA buffer (10 mmol/L Tris and 1mmol/L EDTA, pH = 8.95) for 20 min, 700 W in a microwave. Afterwards sections were gradually cooled down and transferred into coverplates (Thermo Fisher Scientific GmbH, Schwerte, Germany). Sections were rinsed with Tris-buffered saline (TBS) and incubated with undiluted normal goat serum for 30 min to block unspecific binding sites prior to incubation with the primary antibody. Sections were then treated for 1 h with a rabbit polyclonal primary antibody against the major capsid protein p72 of ASFV (diluted in TBS 1:1600). To investigate apoptosis a rabbit antiserum against active caspase-3 (Promega, Madison, WI, USA; 1:200, diluted in TBS) was used. Following washing with TBS sections were incubated with a secondary, biotinylated goat anti-rabbit IgG (Vector Laboratories, Burlingame, CA; diluted in TBS in 1:200) for 30 min. An ABC-kit (Vector; diluted in TBS 1:200, 30 min) was used providing the conjugated horseradish peroxidase which was followed by an incubation with AEC-substrate (DAKO, Hamburg, Germany) for 10 min to visualize positive reactions. The reaction was stopped with ionized water. Finally, sections were counterstained with hematoxylin for 2 min, rinsed in ionized water for 10 min and mounted with Aquatex (Merck).

##### Semi-Quantitative Scoring

A Zeiss AXIO Scope A1 microscope equipped with 2.5×, 10×, 20×, and 40× objectives was used for brightfield microscopial analysis of histological specimen.

Based on the protocol by Galindo-Cardiel et al. [21] sections were semi-quantitatively scored with some adaptations as follows: normal (0), mild (1), moderate (2), severe (3). Scored pathohistological changes of each organ are listed in Table 1. For evaluation, the post-examination masking approach was used [52] to avoid missing subtle or unexpected infection-related changes. An initial evaluation with full access to all study-related information and slides was made to determine the scoring and identify the main lesions. Subsequently, slides were masked and scores were assigned.

#### 4.4.4. Electron Microscopy

To confirm viral infection and to identify ASFV-infected cells, electron microscopy was conducted. Tissue samples were cut (1 mm^3^) and fixed overnight in 2.5% glutaraldehyde buffered in 0.1 M sodium cacodylate pH = 7.2 and 300 mosmol (SERVA Electrophoresis, Heidelberg, Germany). Then, 1% aqueous OsO_4_ was used for post fixation and 2.5% uranyl acetate in ethanol for *en bloc* staining (SERVA Electrophoresis. After a stepwise dehydration in ethanol the samples were cleared in propylene oxide and infiltrated with Glycid Ether 100 (SERVA Electrophoresis. For polymerization, samples were filled in flat embedding molds and incubated for 3 days at 60 °C.

Semithin sections of resin embedded samples were stained with 1% Toluidine Blue O in aqueous sodium borate (Carl Roth) on a hot plate. The POI was trimmed, and prepared ultrathin sections were transferred to formvar coated nickel grids (slot grids; Plano, Wetzlar, Germany). All grids were counterstained with uranyl acetate and lead citrate before examination with a Tecnai Spirit transmission electron microscope (FEI, Eindhoven, The Netherlands) at an accelerating voltage of 80 kV.

#### 4.4.5. Flow Cytometry

Whole blood or leukocytes isolated from the indicated organs were investigated by flow cytometry. Single cell suspensions of tissue samples were prepared as described previously [47]. In total, 50 µL whole blood or 50 µL single cell suspension (approx. 1 × 10^6^ cells) was used for each staining. Incubation steps with monoclonal antibodies were carried out at 4 °C for 15 min in the dark for extracellular staining. Cells were fixed and permeabilized using the True Nuclear Transcription Factor Buffer Set (Biolegend, San Diego, CA, USA) according to the manufacturer’s instructions. For intracellular staining after fixation, cells were incubated at 4 °C for 30 min in the dark. Erythrocytes were lysed before fixation with red cell lysis buffer (1.55 M NH_4_Cl, 100 mM KHCO_3_, 12.7 mM Na_4_EDTA, pH = 7.4, in *Aqua destillata*).

The following antibodies were used in this study: anti-pig CD3ε-APC (clone PPT3, 1:500), anti-pig CD8α-FITC (clone 76-2-11, 1:100), anti-pig CD4-PerCP (clone 74-12-4, 1:100), anti-pig γδ T cell receptor (TCR, IgG2b, clone PPT16, 1:100), anti-pig CD2 (IgG2a, clone MSA4, 1:100), anti-pig CD172α (IgG1, clone 74-22-15; 1:100), anti-pig CD14 (IgG2b, clone MIL2; 1:500), anti-pig CD163-PE (clone 2A10/11; 1:100), and anti-vp72-ASP (IgG2a, clone 18BG3; 1:100). For detection of unconjugated primary antibodies, the following secondary antibodies were used: anti-mouse IgG1-BV421 (clone RMG1-1, 1:400), anti-mouse IgG2b-PE-Cy7 (goat polyclonal, Southern Biotech, 1:400), anti-mouse IgG2a-APC-Cy7 (goat polyclonal, Southern Biotech, 1:250), and anti-mouse IgG2a-APC (goat polyclonal, dianova, 1:500). CD4 was not detectable in one of the control wild boars in trial 2, probably because of a polymorphism in the *CD4* alleles [53]. CD4^+^ and CD4^+^/CD8α^+^ cells from this animal were therefore not included in the analyses.

Dead cells were identified using Zombie Aqua (Biolegend) and excluded from subsequent analyses. Doublets were excluded by FSC-H vs. FSC-A gating. In all tissues, FSC^med^/SSC^low^ were subdivided into CD172^+^/CD14^+^ monocytes/macrophages and CD172^+^/CD14^—^ dendritic cells. FSC^hi^/SSC^hi^/CD172^+^/CD14^+^ cells in the blood were identified as granulocytes. The gating strategy is shown in Appendix A. In the liver, a population of CD172^low^/CD14^+^ cells resembled liver macrophages or Kupffer cells. FSC^low^/SSC^low^/CD3^+^ lymphocytes were differentiated into αβ T cells (CD3^+^/γδ TCR^–^) and γδ T cells (CD3^+^/γδ TCR^+^). Analysis of co-expression of CD4 and CD8α was used to identify subpopulations of αβ T cells. Activated and effector γδ T cells were identified as CD2^+^/CD8α^–^ and CD2^+^/CD8α^+^, respectively. Flow Cytometer BD FACS Canto II with FACS DIVA Software (BD Bioscience, San Jos, CA) and FlowJo^TM^ V10 for Windows (Becton, Dickinson and Company; 2019) were used for all analyses.

#### 4.4.6. Processing of Blood Samples

Serum samples were centrifuged at 2150× *g* for 20 min at 20 °C and EDTA blood was aliquoted and stored at −80 °C until further use. Tissue samples, which were collected during necropsy, were stored at –80 °C. qPCRs were performed after homogenization in 1 mL phosphate-buffered saline (PBS) using a metal bead TissueLyser II (Qiagen GmbH, Hilden, Germany). The supernatant was used for virus isolation (hemadsorption tests).

##### ASFV Antigen and Genome Detection

For rapid antigen detection Lateral Flow device INgezim.ASFCROM Antigeno 11.ASF.K42 (Ingenasa) was used with blood and serum. For qPCR, viral nucleic acid was extracted using the NucleoMag Vet Kit (Machery-Nagel, Düren, Germany) and the KingFisher^®^ extraction platform (Thermo Fisher Scientific, Waltham, MA USA). qPCRs were performed according to the protocols published by King et al. [54] and Tignon et al. [55] with slight modifications. All PCRs were performed using a C1000^TM^ thermal cycler from BIO-RAD (Hercules, CA, USA), with the CFX96^TM^ Real-Time System. Results of both qPCRs were recorded as genome copies per µL.

To detect ASFV in serum, blood and tissue samples a hemadsorption test (HAT) was performed using PBMC-derived macrophages according to slightly modified standard procedures [56]. The samples were tested in quadruplicate.

#### 4.4.7. Statistical Analysis

GraphPad Prism 8 (Graphpad Software Inc., San Diego, CA, USA) was used for statistical analyses and graph creation. Statistically significant differences were investigated by multiple *t*-tests with Holm–Sidak’s correction for multiple comparisons. Statistical significance was defined as *p* < 0.05 and indicated with an asterisk (*).

## Figures and Tables

**Figure 1 pathogens-09-00662-f001:**
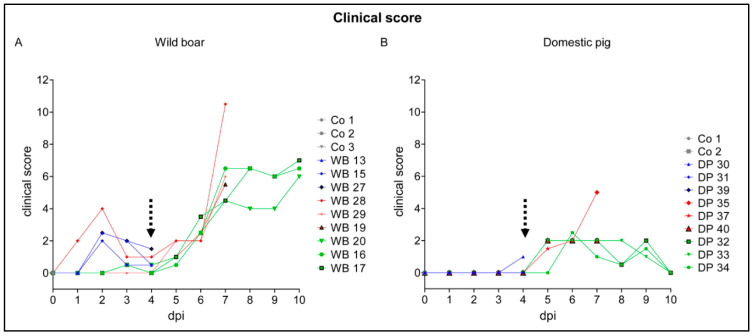
Clinical scores of animals were assessed based on the protocol published by Pietschmann et al. (2015). Clinical scores of wild boar (**A**) and domestic pigs (**B**) were determined daily over the study period. The dotted arrow marks the onset of African swine fever (ASF)-related clinical scores. Co = control, WB = wild boar, DP = domestic pig.

**Figure 2 pathogens-09-00662-f002:**
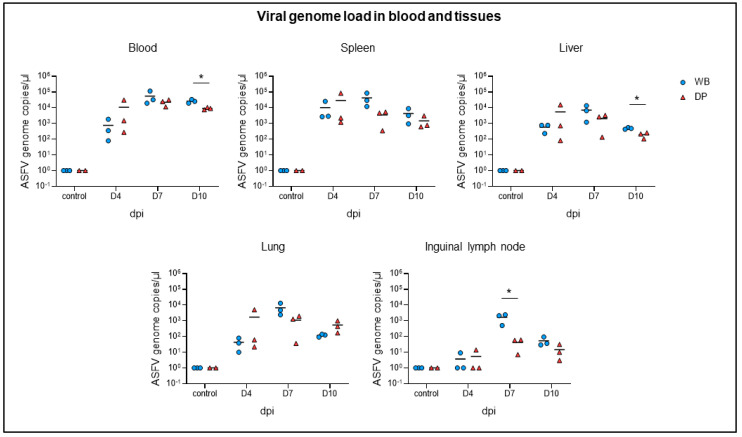
Detection of viral genome in blood and tissues (spleen, liver, lung, inguinal lymph node) of wild boar and domestic pigs by qRT-PCR. WB = wild boar, DP = domestic pig, * *p* < 0.05.

**Figure 3 pathogens-09-00662-f003:**
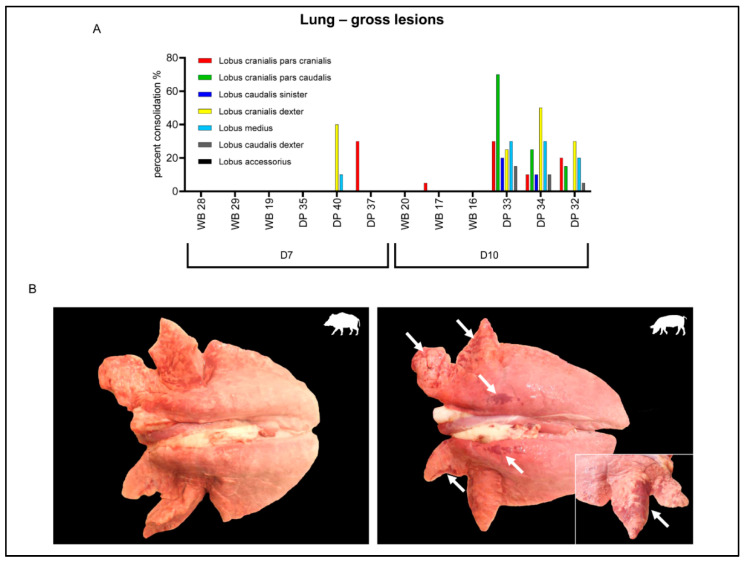
Gross lesions in the lung of wild boar and domestic pigs. (**A**) Macroscopic scoring of percentage pulmonary consolidation in lung lobes occurring at 7 and 10 days post infection (dpi) in wild boar and domestic pigs. (**B**) Lung lesions of WB and DP at day 10 pi. Arrows indicate consolidated areas. WB = wild boar, DP = domestic pig.

**Figure 4 pathogens-09-00662-f004:**
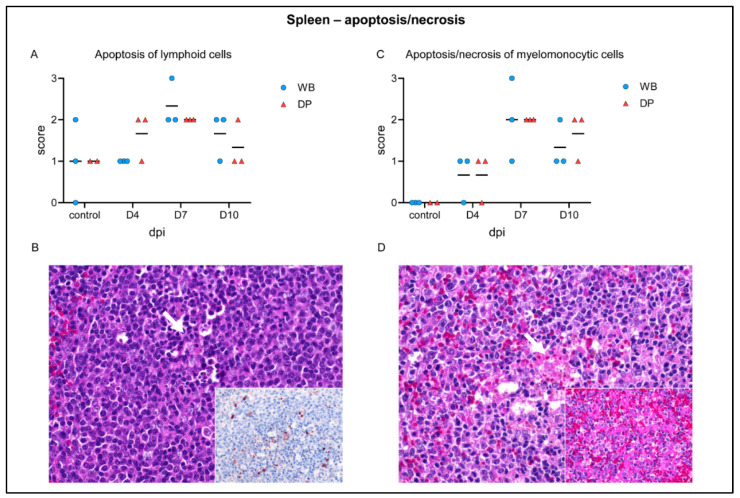
Apoptotic and necrotic changes in the spleen of wild boar and domestic pigs. (**A**) Microscopical scoring of apoptosis of lymphoid cells. (**B**) Pyknosis and karrhyorhexis (arrow) of lymphoid cells and multifocal caspase-3-labelled cells (inset) in a domestic pig at day 4 pi. (**C**) Microscopical scoring of apoptosis/necrosis of myelomonocytic cells. (**D**) Focal coagulative necrosis in the spleen (arrow) and multifocal hyperplastic dendritic cells in a wild boar at day 7 pi. (inset). WB = wild boar, DP = domestic pig, median as horizontal line.

**Figure 5 pathogens-09-00662-f005:**
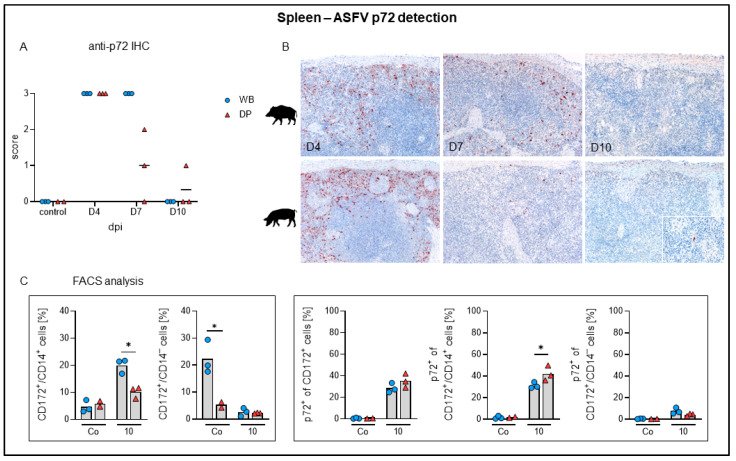
ASFV p72 detection in the spleen of wild boar and domestic pigs. (**A**) Microscopical scoring of p72 positive cells in the spleen. (**B**) Representative sections of anti-p72 immunohistochemistry in wild boar and domestic pigs at days 4, 7, and 10 pi. (**C**) Frequencies of CD172^+^/CD14^+^ monocytes and CD172^+^/CD14^–^ dendritic cells among live leukocytes (left panels) and frequencies of p72^+^ monocytes and p72^+^ dendritic cells (right panels) in the spleens of investigated animals. WB = wild boar, DP = domestic pig, IHC = immunohistochemistry, * *p* < 0.05, median as horizontal line (**A**) or bar (**C**).

**Figure 6 pathogens-09-00662-f006:**
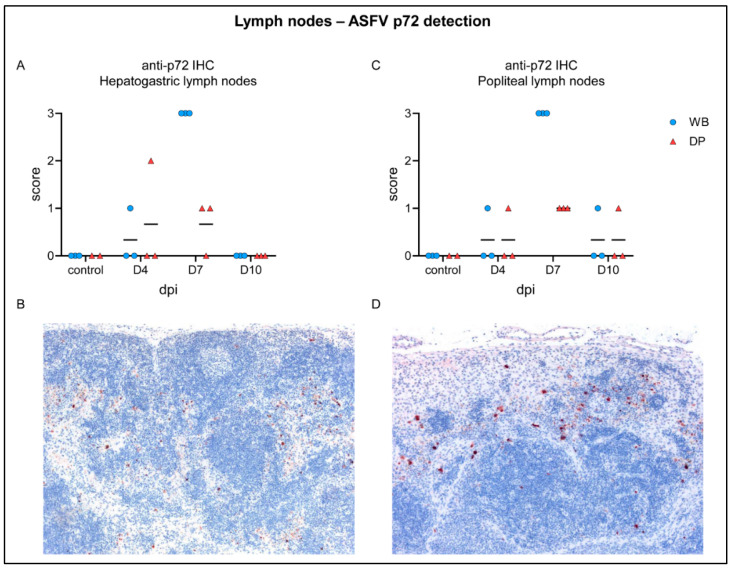
ASFV p72 antigen detection in lymph nodes of wild boar and domestic pigs. (**A**) Microscopical scoring of p72 positive cells in the hepatogastric lymph nodes. (**B**) Section of the hepatogastric lymph node from a wild boar showing abundant p72 postive cells at day 7 pi. (**C**) Microscopical scoring of p72 positive cells in the popliteal lymph node. (**D**) Section of the popliteal lymph node from a WB showing abundant p72 positive cells at day 7 pi. WB = wild boar, DP = domestic pig, median as horizontal line.

**Figure 7 pathogens-09-00662-f007:**
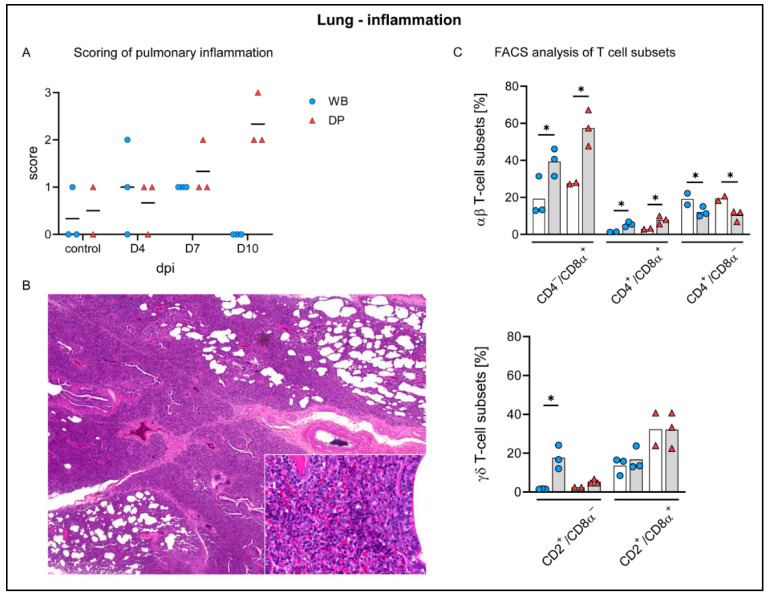
Pulmonary inflammation in wild boar and domestic pigs. (**A**) Microscopical scoring of pulmonary inflammation. (**B**) Section of lung tissue from a domestic pig showing severe interstitial pneumonia with mainly infiltrating lymphocytes and histiocytes (inset) at day 10 pi. (**C**) Frequency of αβ and γδ T-cell subsets in the lung of control animals (white bars) and wild boar and domestic pigs at 10 dpi (grey bars). WB = wild boar, DP = domestic pig, * *p* < 0.05, median as horizontal line (**A**) or bars (**C**).

**Figure 8 pathogens-09-00662-f008:**
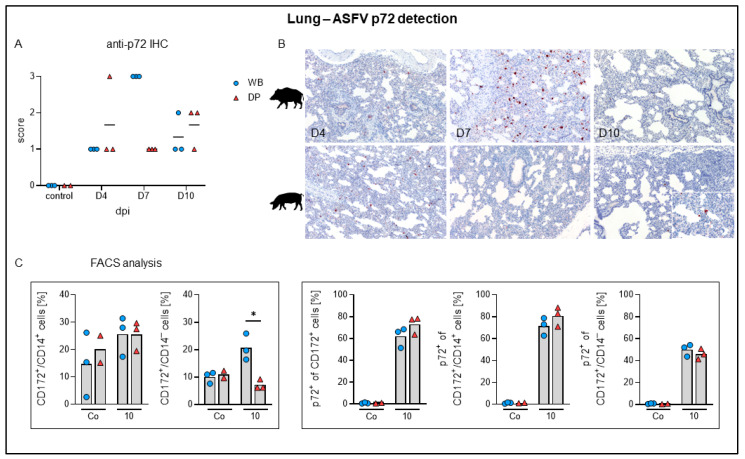
ASFV p72 antigen detection in the lung of wild boar and domestic pigs. (**A**) Microscopical scoring of p72 positive cells in the lung. (**B**) Representative sections of the lung in wild boar and domestic pigs stained with a rabbit anti-p72 serum at days 4, 7, and 10 pi. (**C**) Frequencies of CD172^+^/CD14^+^ monocytes and CD172^+^/CD14^–^ dendritic cells among live leukocytes (left panels) and frequencies of p72^+^ monocytes and p72^+^ dendritic cells (right panels) in the lungs of investigated animals. * *p* < 0.05, median as horizontal line (**A**) or bar (**C**).

**Figure 9 pathogens-09-00662-f009:**
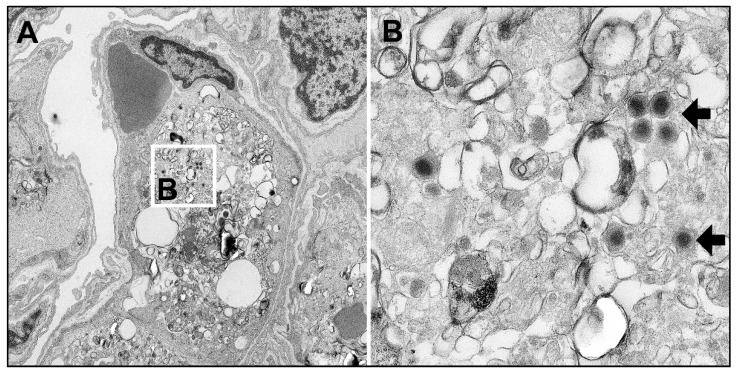
Electron-micrograph of an ASFV-infected pulmonary intravascular macrophage (PIM). (**A**) PIM (center) with multiple virions found in the cytoplasm (white square) (**B**) Detail of figure A showing typical electron-dense virions (arrows).

**Figure 10 pathogens-09-00662-f010:**
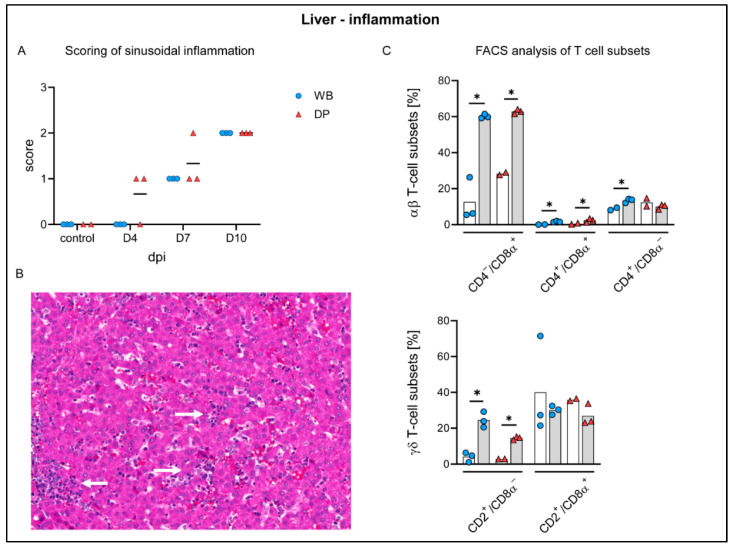
Hepatic inflammatory reaction against ASFV infection in wild boar and domestic pigs. (**A**) Microscopical scoring of sinusoidal inflammation. (**B**) Liver section of a domestic pig showing moderate lymphocytic infiltration (arrow). (**C**) Frequency of αβ and γδ T-cell subsets in the liver of WB and DP in controls (white bars) and 10 dpi (grey bars). WB = wild boar, DP = domestic pig, * *p* < 0.05, median as horizontal line (**A**) or bar (**C**).

**Figure 11 pathogens-09-00662-f011:**
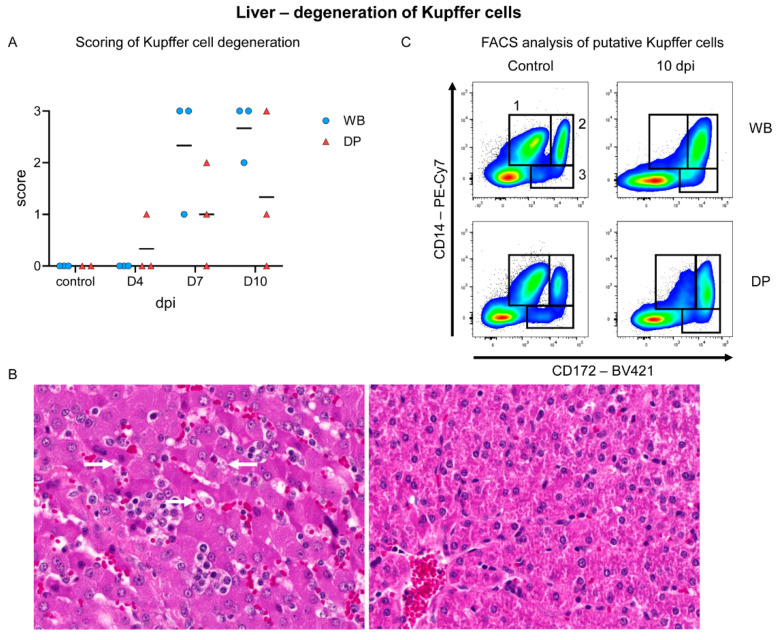
Kupffer cell degeneration in the liver of wild boar and domestic pigs. (**A**) Microscopical scoring of Kupffer cell degeneration in the liver. (**B**) Representative sections of an ASFV infected liver with marked Kupffer cell swelling/degeneration (arrow) in a wild boar 10 dpi (left) and a liver of a control animal (right). (**C**) Identification of myeloid cell populations in the liver by co-expression of CD14 and CD172. (**1**) CD14^+^/CD172^low^, (**2**) CD14^+^/CD172^+^, and (**3**) CD14^–^/CD172^+^ cells were identified as putative liver macrophages or Kupffer cells, monocytes, and dendritic cells, respectively. WB = wild boar, DP = domestic pig, median as horizontal line (**A**).

**Figure 12 pathogens-09-00662-f012:**
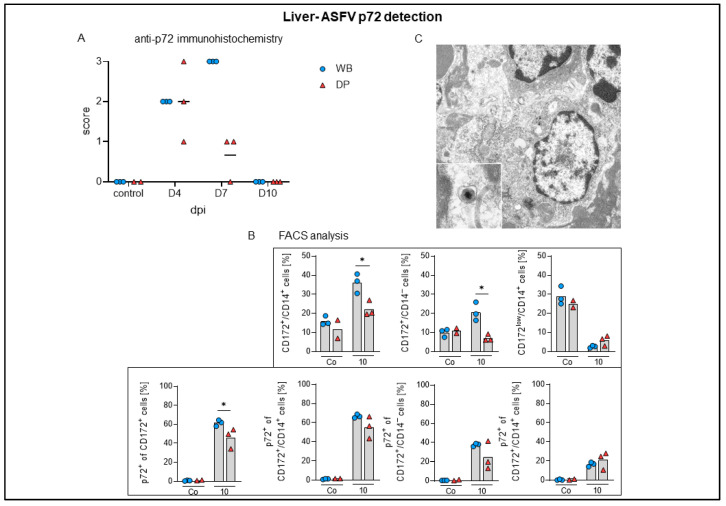
ASFV p72 antigen detection in the liver of wild boar and domestic pigs. (**A**) Microscopical scoring of p72 positive cells in the liver. (**B**) Frequencies of CD172^+^/CD14^+^ monocytes, CD172^+^/CD14^–^ dendritic cells, and CD172^low^/CD14^+^ putative Kupffer cells among live leukocytes (upper panels) and frequencies of p72^+^ monocytes, p72^+^ dendritic cells, and p72^+^ putative Kupffer cells (lower panels) in the liver of investigated animals. (**C**) Electron micrograph of a Kupffer cell with intracytoplasmic ASF virions (inset). WB = wild boar, DP = domestic pig, * *p* < 0.05, median as horizontal line.

**Table 1 pathogens-09-00662-t001:** Histopathological changes in tissues investigated in domestic pigs and wild boar.

Organ	Lesion (Scored with Normal (0), Mild (1), Moderate (2), Severe (3)						
Spleen	Apoptosis of Lymphoid Cells (Syn. Lymphocytolysis)		Apoptosis/Necrosis in the Red Pulp		Hyperplasia/Hypertrophy of Myelomonocytic Cells (i.e., Dendritic Cells, Monocytes, Macrophages)		
Liver with gall bladder	Congestion/hemorrhage	Sinusoidal inflammatory infiltrates		Degeneration/necrosis and loss of Kupffer cells			Hyperplasia of bile duct epithelium
Lymph nodes	Congestion/hemorrhage	Apoptosis of lymphoid cells		Apoptosis/necrosis, perifollicular cortex and paracortex	Presence of tingible body macrophages		Hyperplasia of lymphocytes
Palatine tonsil	Congestion/hemorrhage	Apoptosis of lymphoid cells		Apoptosis/necrosis, perifollicular cells		Crypt abscessation	
Lung	Congestion/hemorrhage	Edema	Inflammation	Hyperplasia/hypertrophy of alveolar and interstitial monocytes/macrophages			
Kidney	Congestion/hemorrhage			Inflammatory infiltrates			
Brain	Inflammatory infiltrates			Neuronal degeneration	Gliosis		
Bone marrow	Necrosis						

Immunohistochemically stained sections were scored for viral antigen distribution and scored on a 0–3 scale. The most affected area per sample sections was scored with no antigen (0), < 1–3 positive cells (1), 4–15 cells (2), and >16 cells (3) per high (40×) power field. Cells showing fine granular cytoplasmic labelling were considered positive; chromogen aggregations without cellular association were not counted.

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
