# Peer review of "Comparative Pathology of Domestic Pigs and Wild Boar Infected with the Moderately Virulent African Swine Fever Virus Strain “Estonia 2014”"

_pathogens, 2020, doi:10.3390/pathogens9080662_

Round 1
Reviewer 1 Report
This is a well designed study and well written manuscript that provided interesting comparative information about the early pathogenesis of ‘Estonia ASFV’ in wild boar and domestic swine.
In this reviewer’s opinion, the following comment that should be addressed prior to publication.
1)Within the manuscript, please discuss the effects of potential genotypic difference between domestic swine and wild boars and how this may affect the susceptibility of disease. Its likely both the virulence of ASF virus and possible innate resistance to disease (ie domestic swine) influenced the manifestation of disease within the study.
2) Were the wild boar used in the study progeny from a single breeding pair, or did the boars have different genetic backgrounds. The heterogeneity of the genetic lines could affect the results provided, as such please address this within discussion.
3) Antigen specific neutralizing antibody were not evaluated in this study and the presence of antibody could impact the kinetics of viral load with in blood and tissue. Please address the how neutralizing antibodies may impact could affect the manifestation, especially since domestic appeared to recover by day 10 post infection, while wild boar had moderate disease.
4) How was the amount of virus employed for challenged study (HAU, x105.25) determined.
5) The gross pathology, histopathology and immunohistochemistry were ‘semi-quantitatively assessed. Was the evaluators of the tissue and slides 'blinded' to the experimental design, please address this comment in the methods section
Reviewer 2 Report
Sehl at al. present a comprehensive pathological study comparing infection with a moderate virulent African swine fever virus (ASFV) isolate between domestic pigs and wild boars. This kind of study is rarely undertaken and provides new information regarding disease pathogenesis in these subspecies. The manuscript is well written. However, some aspects could be modified to help the reader interpret the data. Specifically:
- Line 105: This sentence is not clear. Fever was considered only when temperatures were above 40 but it is then written “lasting until day 10, ranging from 39.2°C up to 40.6°C”. Please give details of how many animals still had high temperatures and at each day. Alternatively, a graph showing the temperatures would be very helpful.
- Lines 129 -137 and 610-619: The gating strategy should be shown as supplementary material.
- Lines 135-137. The percentage of p72+ granulocytes was extremely high (up to 98% at 10 pi). Since these cells are not the natural host for viral replication, this finding should be further investigated. Key data missing is the mean fluorescence intensity (MFI) of these positive cells compared to positive monocytes. It should be expected that active viral replication and phagocytosis of viral particles would result in different MFI.
- Line 401: Perhaps a better description would be “virus attachment to erythrocytes” rather than “infected erythrocytes”.
Minor points:
- Line 77: lesions instead of lesion
- Line 621: This sentence seems to indicate that EDTA blood was also centrifuged?
Author Response
Please see the attachment for Reviewer 2. The rebuttal has been drafted with all changes in one letter.

Round 2
Reviewer 1 Report
The final version of the manuscript is very good and all comments were adequately addressed by the authors